# TwinShield: Secure Foundation Model Execution by Unifying TEEs and Crypto-protected Accelerators

## Abstract

Recent advances in Transformer-based foundation models (FMs) have driven significant developments across diverse AI tasks, facilitating their deployment in security-sensitive domains. Despite their capabilities, FMs impose substantial inference costs, driving reliance on third-party cloud infrastructure equipped with high-performance computation resources. However, these cloud platforms cannot be fully trusted and remain vulnerable to data breaches, introducing dual confidentiality challenges: protecting user data from exposure and safeguarding models against unauthorized access. Mainstream protection mechanisms leverage trusted execution environments (TEEs), where confidentiality and integrity are enforced through hardware-based isolation, encryption, and integrity verification. But executing inference entirely within TEEs incurs a significant overhead, which is further exacerbated in large-scale FMs. Recent studies have proposed schemes that combine TEEs with untrusted accelerators (e.g., GPUs) to offload partial inference operations. However, prior offloading schemes cannot solve dual confidentiality challenges in FM inference, since operations such as `Attention` depend on dynamic operands that prevent secure precomputation and must remain within TEEs. Moreover, the communication overhead between TEEs and accelerators grows dramatically with model scale, constituting a new system design challenge for FMs. To address these challenges, we propose TwinShield, a framework that enables secure inference of Transformer-based FMs in heterogeneous TEE–accelerator systems with dual protection for both model and data. TwinShield improves efficiency through *protocol-level* outsourcing, which securely offloads the majority of operations to accelerators, and enhances throughput via a *system-level* design that overlaps TEE preparation, communication, and accelerator execution. Our evaluation on representative LLMs and VLMs shows that TwinShield offloads about $87\%$ of computations to accelerators and achieves $3.3\times$–$5.1\times$ speedups over baselines. The code is publicly available at https://anonymous.4open.science/r/Twinshield.

## 1 Introduction

With the rapid advances in Transformer architectures (Vaswani et al., 2017), they have been widely applied in domains such as computer vision (Dosovitskiy et al., 2020) and natural language processing (Devlin et al., 2018). Building on this architecture, large-scale foundation models (FMs) such as LLaMA and Qwen have emerged. Benefiting from their remarkable capabilities, FMs are becoming increasingly popular and are being deployed in many critical scenarios. However, these capabilities are primarily driven by the enormous parameter sizes of such models, which consequently impose significant computational demands. To address the challenges of model size and deployment complexity, cloud-based Foundation Model-as-a-Service (FMaaS)[1] has become a widely adopted paradigm, enabling model owner to provide state-of-the-art FMs as inference services to end users in a cost-effective manner.

---

[1] https://builder.aws.com/building-a-foundation-model-as-a-service-fmaas-on-aws

In the FMaaS, input data provided by clients, such as personal health information (e.g., sleep patterns, pulse, heart rate) and financial records, is highly sensitive. Meanwhile, model providers delegate the hosting and execution of their FMs to the cloud, which constitutes valuable intellectual property, since developing them requires enormous investments in data collection, domain expertise, and computational resources for training. Despite leveraging the cloud's powerful computation resources, remote execution cannot be fully trusted, as adversaries may exploit privileged system software (Pahima, 2022) or hardware vulnerabilities (Tung, 2021) to compromise privacy and computation integrity. Therefore, guaranteeing the *confidentiality* of both client inputs and provider models, as well as the *integrity* of inference, is imperative for FMaaS.

Trusted Execution Environments (TEEs), such as Intel SGX, provide a trusted environment to safeguard the privacy and integrity of sensitive computations. In systems with TEEs, the CPU is treated as the root of trust. The processor shields individual secure enclaves from privileged system software attacks via hardware-enforced isolation. Furthermore, counter-mode encryption and integrity tree-based data verification are performed by the TEE hardware to protect against breaches and tampering with enclave off-chip data. Accordingly, prior studies have investigated the use of TEEs for secure

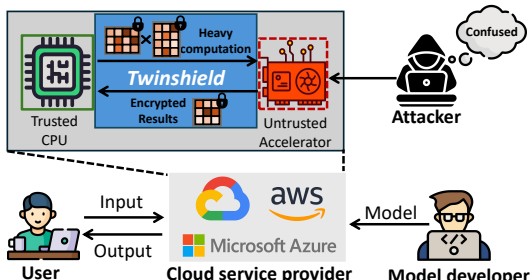

Figure 1: Overview of our trusted foundation models (FMs) inference, TwinShield.

machine learning inference. For instance, Hanzlik et al. (2021) proposed to store ML models in the secure enclave and perform inference completely in TEEs, hence protecting computation integrity and the confidentiality of all data. Unfortunately, deployment of the entire model inside TEEs introduces extremely high overhead due to the limited resources of TEEs. Recent advances in TEE-based accelerators (e.g., NVIDIA H100 confidential mode) attempt to mitigate this issue, but they remain vendor-specific, technically restrictive, and provide weaker guarantees than CPU-based TEEs. Subsequent works (Tramer & Boneh, 2018; Hashemi et al., 2021; Sun et al., 2023; Shen et al., 2022) attempt to improve the performance of TEE-based model inference by *outsourcing* heavy computations from TEEs to an untrusted external accelerator (e.g., GPUs, FPGAs and ASICs), and *verifying* the computation integrity inside the enclave. While the aforementioned secure outsourcing techniques enhance the system efficiency of TEE-only methods, they struggle to outsource sufficient computations of Transformer-based FMs to untrusted accelerators from trusted TEEs. The challenges are summarized as follows:

*(I) Confidential Attention Computation:* Traditional schemes for non-Transformer models rely on additive secret sharing to outsource linear operations $W \cdot x$ by sending the masked input $x + r$ to untrusted accelerator for $W \cdot (x + r)$ and recovering the result by subtracting the precomputable $W \cdot r$. In contrast, `Attention` involves computations such as $q \cdot k$ and $\texttt{softmax}(qk) \cdot v$, where both operands are generated at runtime. This property precludes precomputation and renders existing methods inapplicable. Moreover, prior studies in cloud settings (Tramer & Boneh, 2018; Hashemi et al., 2021) focus solely on input protection while assuming the model resides with the server, leaving it unprotected. Conversely, on-device approaches (Shen et al., 2022; Sun et al., 2023) focus on model privacy but treat user inputs as local and leave them unprotected. We argue that both the model and inputs must be protected simultaneously, a setting substantially more complex than safeguarding either alone. Achieving this dual protection requires obfuscating both components before outsourcing any operation to untrusted accelerators, necessitating a redesign of secure computation algorithms.

*(II) Significant Communication:* Foundation models contain billions of parameters, making accelerator–TEE communication non-negligible and increasingly costly. For example, outsourcing a single layer of LLaMA-8B can incur 3.38 GB of bidirectional data transfer. While prior works designed for small models such as CNNs tolerate this overhead, they become inefficient when applied to FMs. This scalability gap underscores the need for new system designs that mitigate the substantial communication inherent in outsourcing large-scale FMs.

To address these challenges, we propose TwinShield (as shown in Figure 1), a framework for confidential and verifiable inference on Transformer-based FMs. The model developer deploys the model on the cloud to process the client input. TwinShield's protocol enables most computations

to run on accelerators while ensuring data confidentiality and computation integrity. For Challenge (I), we design a confidentiality-guaranteed outsourcing protocol, OutMult, which consists of two components: OutAttnMult for `Attention` computations and OutLinearMult for weight–input multiplications, both ensuring protection of model and input. For Challenge (II), our key insight is that attention heads can be computed independently, which allows their workloads to be decomposed into smaller parallel tasks. By pipelining TEE computation, data transfers, and accelerator computation across different heads, we effectively utilize the idle time. The proposed design, i.e., OutPipe, overlaps communication and computation, thereby improving hardware utilization and increasing throughput by $52.4\%$. Through extensive experiments on various FMs, such as large language models (LLMs) and vision language models (VLMs), we show that $\mathrm{TwinShield}$ achieves substantial throughput improvements ranging from $3.3\times$ to $5.1\times$ for private verifiable inferences, without sacrificing accuracy.

## 2 THREAT MODEL

We consider a cloud-based Foundation Model-as-a-Service (FMaaS) scenario with three parties: the model developer, the cloud service provider, and the model user, as illustrated in Figure 1. The model developer trains and then deploys a foundation model $f : X \to Y$ on the cloud service provider. The user queries the model through the cloud service. The cloud service provider is equipped with a trusted CPU TEE (e.g., Intel SGX) that serves as the root of trust, and an untrusted accelerator (e.g., GPU) that performs heavy computations but is not fully trustworthy.

An ideal protection scheme should satisfy the following security properties:

- **Data Privacy:** The cloud server cannot learn any information about the input $x$.

- **Model Privacy:** The cloud server cannot learn any information about the model $F$.

- **t-Integrity:** The probability that a user accepts an incorrect output $\tilde{y} \neq F(x)$ from the cloud server without aborting is less than $t$.

We treat the CPU TEE as the secure and reliable root of trust (Tramer & Boneh, 2018; Hashemi et al., 2021), which can be verified through remote attestation. Our goal is to extend these guarantees to outsourced computations executed on the untrusted accelerator.

We note that Intel SGX and other TEEs have been shown vulnerable to side-channel attacks and denial-of-service attacks (Van Bulck et al., 2018; Van Schaik et al., 2019). These attacks have been extensively studied, and a wide range of defense mechanisms have been proposed, including constant-time implementations, oblivious memory primitives, and obfuscation techniques that conceal both code and data access patterns (Brasser et al., 2019; Lou et al., 2021; Ahmad et al., 2019; Wichelmann et al., 2024). Such defenses are orthogonal to the focus of this work, which addresses different aspects of secure computation.

## 3 BACKGROUND AND RELATED WORK

### 3.1 TRANSFORMER-BASED FOUNDATION MODELS

Transformer architectures (Vaswani et al., 2017) have become the backbone of modern AI, achieving state-of-the-art performance in natural language processing (Myers et al., 2024), computer vision, and multi-modal tasks (Awais et al., 2025). Building on this architecture, large-scale foundation models such as LLaMA (Touvron et al., 2023), Qwen (Yang et al., 2025) and Phi (Abdin et al., 2024) have emerged, with billions of parameters and pretraining on massive corpora. These models demonstrate strong generalization and transferability, enabling deployment across diverse applications, including dialogue systems, code generation, healthcare, and finance. The `Attention` is the core module of the Transformer architecture, which can be formulated as:

$$X_l = \mathrm{Attention}(Q, K, V) = \mathrm{SoftMax}(QK^T/\sqrt{d_h})V$$

where $W$ are the model parameters with a size of $d_h \times d_h$, and the query, key, and value are computed via $Q = X_{l-1}W_l^Q$, $K = X_{l-1}W_l^K$, and $V = X_{l-1}W_l^V$.

Due to their massive scale, FMs are typically deployed in the cloud, where client inputs may contain sensitive information and model parameters represent valuable intellectual property. This dual confidentiality requirement necessitates protecting both user data and proprietary models during inference. To address this challenge, we propose TwinShield, a framework that enables efficient and secure execution of Transformer-based foundation models with dual protection guarantees.

## 3.2 TRUSTED EXECUTION ENVIRONMENTS (TEES)

Trusted Execution Environments (TEEs) such as Intel SGX provide secure enclaves that guarantee confidentiality and integrity of computations by isolating code and data from the rest of the system, including the operating system and hypervisor. These hardware-based protections have motivated research into running deep learning inference inside TEEs to protect sensitive user data and proprietary models. However, the high computational and memory demands of modern foundation models make TEE inference inefficient, motivating the use of accelerators such as GPUs to improve performance.

**Limitations of Accelerators with TEEs.** Most current AI infrastructures and cloud platforms lack TEE-based accelerators, as extending accelerator support remains vendor-specific, technically challenging, and fraught with unresolved security concerns. Recent studies show that even NVIDIA H100 GPUs with confidential mode fall short of the security guarantees offered by CPU-based TEEs, underscoring the need for further refinement of secure accelerator designs (Gu et al., 2025; Mohan et al., 2024). In addition, many data centers still rely on legacy GPUs, such as A100s and V100s, making it necessary to explore how these widely deployed accelerators can perform confidential computing. A common approach is to treat the CPU TEE as the root of trust and offload heavy linear operations to untrusted accelerators through controlled interfaces. Yet, existing protocols cannot simultaneously protect both inputs and model weights, and have been demonstrated only on small-scale models such as CNNs. Extending secure support to key primitives in Transformer-based FMs, particularly the attention mechanism, remains a pressing and unresolved challenge.

## 3.3 RELATED WORK

We compare TwinShield with several lines of related work in Table 1. The first category executes all computations inside TEEs, with representative work such as Occlumency (Lee et al., 2019). While this approach guarantees strong security, it suffers from significant efficiency loss due to the limited computational resources available within TEEs. The second category focuses on protecting user input data while outsourcing linear operations to untrusted hardware. Representative examples include Slalom (Tramer & Boneh, 2018) and DarKnight (Hashemi et al., 2021), which assume that the model belongs to the cloud provider and therefore do not

Table 1: OutSrc. stands for outsource and Cld Infer. for cloud inference. ● denotes supported, ○ denotes not supported, ⋆ denotes the user input privacy is protected by on-device setting.

| Method | Model Privacy | Input Privacy | Inference Integrity | Linear OutSrc. | Attn OutSrc. | Comm. Optim. | FMs Cld Infer. |
|---|---|---|---|---|---|---|---|
| Occlumency | ● | ● | ● | ○ | ○ | ○ | ○ |
| Slalom | ○ | ● | ● | ● | ○ | ○ | ○ |
| DarKnight | ○ | ● | ● | ● | ○ | ○ | ○ |
| SOTER | ● | ⋆ | ● | ● | ○ | ○ | ○ |
| ShadowNet | ● | ⋆ | ● | ● | ○ | ○ | ○ |
| NNSplitter | ● | ○ | ○ | ● | ○ | ○ | ○ |
| TSDP | ● | ⋆ | ○ | ● | ○ | ○ | ○ |
| GroupCover | ● | ⋆ | ● | ● | ○ | ○ | ○ |
| TwinShield | ● | ● | ● | ● | ● | ● | ● |

address model confidentiality. A third category of work, mainly in on-device scenarios (e.g., SOTER (Shen et al., 2022), ShadowNet (Sun et al., 2023), and others (Zhou et al., 2023; Zhang et al., 2023; Liu et al., 2023; Zhang et al., 2022; Sun et al., 2025; Zhang et al., 2024)), shifts the focus to model privacy while assuming user inputs remain local and thus unprotected. Since these protocols do not protect input privacy during computation, they are not applicable to cloud FM inference setting.

Our work, TwinShield, is the first framework designed for dual protection of model and input in the cloud setting. Moreover, it is the first to efficiently support large-scale FM inference with both secure outsourcing of attention computations and optimized TEE-accelerator communication, two challenges unique to FM inference that prior approaches did not address.

# 4 TwinShield DESIGN

Figure 2 provides an overview of TwinShield, our proposed framework for secure inference. Since the majority of the computation in Transformer-based Foundation Models comes from large-scale matrix multiplications (Hoffmann et al., 2022), we propose OutMult which includes OutAttnMult (in Sec. 4.1) and OutLinearMult (in Sec. 4.2). Then, we propose OutPipe in Sec. 4.2 to overlap computation and communication, further improving throughput. By outsourcing these bottlenecks, we can significantly improve overall efficiency and enable secure Transformer inference at scale.

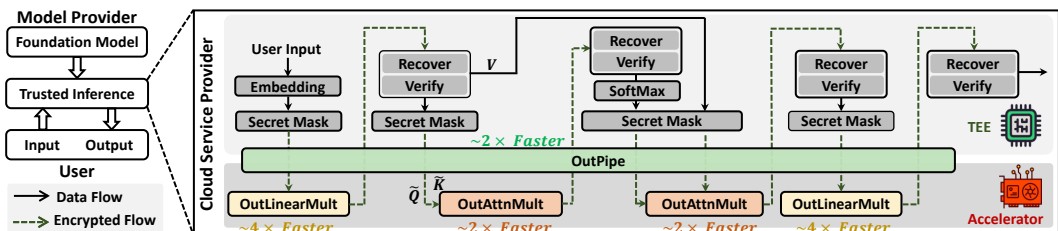

Figure 2: Overview of TwinShield. The model developer deploys a foundation model to the cloud service provider, which hosts a trusted CPU TEE and an untrusted accelerator. The user submits input to the trusted TEE, which masks sensitive inputs and model parameters before outsourcing heavy computations to the untrusted accelerator. We propose three secure outsourcing protocols: OutLinearMult for linear layers, OutAttnMult for attention operations, and OutPipe for pipelined communication and computation.

## 4.1 OUTSOURCE ATTENTION OPERATION: OutAttnMult

Unlike linear operations between weights and inputs, `Attention` in FMs involves two variable operands, namely the multiplications between $Q$ and $K^T$, and between `SoftMax`$(QK^T)$ and $V$. This variability prevents TEEs from precomputing masked products, as done in prior work (Tramer & Boneh, 2018; Hashemi et al., 2021; Sun et al., 2023), since the operands are not known before inference. Consider the multiplication $QK^T$: the TEE masks $Q$ with $R_Q$ and $K^T$ with $R_K^T$, and outsources $(Q + R_Q)(K^T + R_K^T)$ to the accelerator. The result expands to

$$QK^T + R_Q K^T + Q R_K^T + R_Q R_K^T$$

To recover $QK^T$, the TEE must subtract the additional terms. Among them, only $R_Q R_K^T$ is precomputable, since the other terms depend on the unknown matrices $Q$ or $K^T$.

We observe that the un-precomputable terms, $Q R_K^T$ and $R_Q K^T$, each involves one predetermined mask, which seems to allow outsourcing through precomputation. For example, the TEE could outsource $(Q + R_Q) \cdot R_K^T$ and then recover $Q R_K^T$ by subtracting the precomputed $R_Q R_K^T$. However, this naïve strategy compromises security: exposing $R_K^T$ enables the accelerator adversary to infer $K^T$ by simply subtracting it from the masked value $K^T + R_K^T$ in the first outsourcing round.

To prevent this risk, we propose a *Scale-then-Permute* strategy. Rather than exposing $R_K^T$ directly, the TEE embeds $R_K^T B$ into the masked matrix $K^T + R_K^T$ with a column-wise permutation, where $B$ is a scalar matrix. This achieves two goals: (i) it hides the distinction between $K^T + R_K^T$ and $R_K^T B$, so an attacker cannot recover $K^T$ by simple subtraction without knowing the secret $B$; and (ii) it allows the accelerator to compute $(Q + R_Q)(K^T + R_K^T)$ and $(Q + R_Q)R_K^T B$ in a single round, avoiding extra communication. The TEE en restores $Q R_K^T$ by applying the inverse permutation and scaling with $B^{-1}$, and subtracting $R_Q R_K^T$. An analogous construction applies symmetrically to $R_Q$.

**Workflow and Complexity Analysis.** The OutAttnMult protocol, illustrated in Figure 3, proceeds in two phases. In the *offline* phase, the TEE samples masks $R_Q$ and $R_K$, and generates their scaled variants using diagonal matrices $A$ and $B$. In the *online* phase, the TEE embeds and permutes the masked inputs to construct $\widetilde{Q} \in \mathbb{F}^{2m \times n}$ and $\widetilde{K}^T \in \mathbb{F}^{n \times 2p}$ with additions and permutations of cost $O(mn + np)$. The accelerator then performs the dominant multiplication $\widetilde{Q}\widetilde{K}^T$. Finally, the TEE recovers the result using four scalings and five additions, also bounded by $O(mn + np)$. In summary, OutAttnMult reduces the TEE workload from $\mathcal{O}(mnp)$ multiplications to only $\mathcal{O}(mn + np)$ lightweight scalar operations, while offloading the $\mathcal{O}(mnp)$ multiplication to the accelerator.

**Security Analysis.** We model the accelerator as an adversary $\mathcal{A}$ interacting with a TEE oracle $\mathcal{O}$. For each query, $\mathcal{O}$ returns $\widetilde{Q} = \texttt{perm}(Q+R_Q, AR_Q; \lambda_Q)$ and $\widetilde{K^T} = \texttt{perm}(K^T+R_K^T, R_K^T B; \lambda_K)$, where masks, scalings, and permutations are secret. Since $Q + R_Q$ forms a one-time pad (Bellare & Rogaway, 2001), its distribution is indistinguishable from $AR_Q$ in the adversary's view, while the scale-then-permuted embedding of $R_K^T B$ prevents subtraction attacks on $K^T$. Thus, no adversary running in probabilistic polynomial time (PPT) can recover $Q$ with non-negligible advantage. The security level can be estimated as $\log_2\left(\binom{2m}{m} m! |\mathbb{F}|^m\right)$ (scalar 8-bit, i.e., $|\mathbb{F}| = 2^\ell$ with $\ell = 8$); for a typical input length $m = 512$, this is $\approx 8{,}990$ bits, far exceeding 128/256-bit security; further details appear in Appendix D.

Figure 3: Protocol of OutAttnMult.

Figure 4: Protocol of OutLinearMult.

## 4.2 OUTSOURCE LINEAR OPERATIONS: OutLinearMult

Existing outsourcing schemes such as Slalom (Tramer & Boneh, 2018) protect only the input in linear computations $Y = WX$: the TEE blinds $X$ with a random mask $R_X$ and precomputes $WR_X$ offline before outsourcing $(X + R_X)W$. This ensures input privacy but leaves the weight matrix $W$ exposed, i.e., no dual protection. A natural extension is to also mask the weights by $W + R_W$, so the accelerator computes $(W + R_W)(X + R_X)$, and the TEE recovers $WX$ by subtracting $WR_X$, $R_W X$, and $R_W R_X$. However, as in attention outsourcing, $R_W X$ cannot be precomputed since $X$ is unknown before inference. One option is to reuse the OutAttnMult protocol, but the linear case is simpler because $W$ is known in advance. This allows us to design a more lightweight protocol, OutLinearMult (in Figure 4). Here, the TEE integrates $R_W$ with $W + R_W$ via a *Scale-then-Permute* strategy, analogous to handling $R_Q$ and $Q + R_Q$ in OutAttnMult while precomputable terms such as $WR_X$ are handled offline.

**Workflow and Complexity Analysis.** As shown in Figure 4, the protocol proceeds in two phases. In the *offline* phase, the TEE samples a weight mask $R_W \in \mathbb{F}^{m \times n}$ and an input mask $R_X \in \mathbb{F}^{n \times p}$, selects a diagonal scalar matrix $C \in \mathbb{F}^{m \times m}$, and precomputes $WR_X$ and $CR_W$. In the *online* phase, the TEE masks and permutes the inputs to obtain $\widetilde{W} \in \mathbb{F}^{2m \times n}$ and $\widetilde{X} \in \mathbb{F}^{n \times p}$, and outsources them to the accelerator. The accelerator then performs the dominant multiplication $\widetilde{W}\widetilde{X}$ with complexity $\mathcal{O}(mnp)$. Finally, in the recovery stage, the TEE applies one scaling and a few additions, with total cost $O(mp)$. Altogether, OutLinearMult reduces the TEE workload from $\mathcal{O}(mnp)$ multiplications in vanilla secure linear computation to only $\mathcal{O}(mn + mp)$ lightweight operations, while offloading the main $\mathcal{O}(mnp)$ cost to the accelerator.

**Security Analysis.** Similar to the protocol of OutAttnMult, we defer the details to the Appendix D.

**t-Integrity Guarantee.** Following Slalom (Tramer & Boneh, 2018), TwinShield verifies outsourced multiplications using Freivalds' algorithm (Freivalds, 1977). Given matrices $A, B$ and a candidate result $C$, the TEE samples a random vector $s$ and checks whether $Cs = A(Bs)$. If

$C \neq AB$, the probability of passing is at most $1/|\mathbb{F}|$, which drops to $t = (1/|\mathbb{F}|)^k$ after $k$ repetitions. Each check costs only $O(n^2)$, much cheaper than recomputing the $O(n^3)$ product inside the TEE, thus providing efficient and tunable t-Integrity for outsourced linear operations.

### 4.3 OUTSOURCE COMMUNICATION-COMPUTATION OPTIMIZATION: OutPipe

Existing TEE offloading schemes adopt a *serial workflow*: inputs are transferred to the accelerator, computation is performed, and results are copied back to TEE for post-processing.

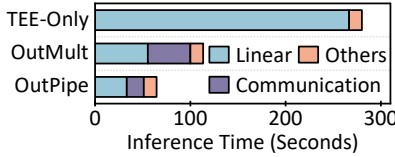

This design was acceptable for small models such as CNNs, however, foundation models with billions of parameters impose significant communication overhead. As shown in Figure 5, after outsourcing linear and attention computations via OutMult, the bottleneck shifts from computation to communication. This arises because in the *serial workflow*, the TEE must wait for all accelerator results (and vice versa) before proceeding, leaving both sides idle during transfers. As demonstrated in Figure 6 (upper), the TEE remains idle during data transfers and accelerator computation.

Figure 5: Inference Breakdown.

To address this challenge, we propose OutPipe, a pipelined workflow that overlaps communication and computation to eliminate the idle time of the serial design. The key observation is that workloads such as multi-head attention exhibit independence across heads, which we group into compute blocks. Each block proceeds through four pipeline stages: preparation inside the TEE, data transfer from TEE to accelerator, computation on the accelerator, and data transfer back to the TEE. This design leverages accelerators that support concurrent copy-and-compute (NVIDIA, 2025; AMD, 2025). To coordinate the TEE and the accelerator, we organize their communication through a shared ring buffer divided into slots. Each slot holds one compute block together with a state flag (READY or DONE). The TEE fills a free slot and marks it READY, while the accelerator processes ready slots and marks them DONE once finished. This mechanism decouples the two sides: the TEE can continue preparing the next block without waiting, and the accelerator can continuously fetch new work. As shown in Figure 6 (bottom), the pipelined design achieves fine-grained overlap across TEE pre-processing, communication, accelerator computation, and TEE post-processing. Once the TEE finishes pre-processing the first block, it immediately starts

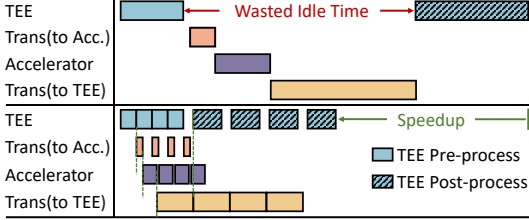

Figure 6: Comparison of baseline (upper) and proposed OutPipe (bottom).

transferring it to the accelerator while continuing to prepare the next block. Once the first block arrives, the accelerator begins computation in parallel with the ongoing transfer of the second block. When the accelerator completes a block, the results are directly transferred back to the TEE while the accelerator proceeds with the next computation. This staged handoff ensures that all components (TEE, communication channels, and accelerator) remain active simultaneously, thereby maximizing utilization and throughput.

## 5 EXPERIMENTAL METHODOLOGY

In this section, we introduce the experimental methodology.

**Models.** We evaluate our approach on four models from three LLM families: LLaMA3 (3B and 8B), Qwen3 (14B), and Phi-4 (14B). In addition to these LLMs, we also include two vision-language models, Qwen2.5-VL (7B) and Pixtral (12B) to cover multimodal tasks.

**System Setup and Implementation.** We conducted the TwinShield implementation on a server equipped with an Intel(R) Xeon(R) Gold 6342 CPU running at 2.8GHz and 512GB DRAM, together with an NVIDIA A40 GPU with 48GB VRAM. TEE enclave is built on the Gramine LibOS, which runs unmodified applications inside Intel SGX. Communication between TEE and the accelerator is

enabled via a shared memory region, with EDMM (Enclave Dynamic Memory Management) activated to support dynamic enclave resizing and thread management. Since our threat model excludes denial-of-service attacks, we assume reliable communication between the CPU and the accelerator. Furthermore, our matrix computation and model inference framework builds upon `ggml` and `llama.cpp`, which provide an efficient and lightweight large language model inference pipeline.

**Quantization.** TwinShield adopts a quantization strategy for both activations and model weights, drawing on the approaches of Slalom Tramer & Boneh (2018) and DarKnight Hashemi et al. (2021). Specifically, it converts values from floating-point to fixed-point by selecting a fractional bit number $l$ (we use $l = 8$ in our implementation), scaling values by $2^l$, and rounding to integers. For negative values, a correction $p$ is applied to adjust them into the field $\mathbb{Z}_p$, where the prime is chosen as $p = 2^{24} - 3$. The TEEs then outsource the subsequent computations to the GPUs, and later dequantize the results to recover the original values.

# 6 EXPERIMENTAL RESULTS

## 6.1 END-TO-END PERFORMANCE

**Comparison with baseline methods.** We evaluate the proposed TwinShield and TEE-only baseline (Hanzlik et al., 2021) across four large-scale foundation models, including LLaMA3-3B, LLaMA3-8B, Qwen3-14B and Phi4-14B. To ensure fairness, both methods are tested under the same setting, using identical models and adopting the same quantization scheme. As shown in Figure 7, TwinShield consistently outperforms the TEE-only baseline across different pre-filling token lengths. For LLaMA3-3B, TwinShield achieves a $3.33\times$ speedup at 2,048 tokens, while the gains increase to $4.13\times$ for LLaMA3-8B, $5.03\times$ for Qwen3-14B, and $5.11\times$ for Phi4-14B. These results demonstrate that the benefit of TwinShield scales with model size, effectively reducing the overhead of secure inference from hundreds of seconds to a fraction of the baseline.

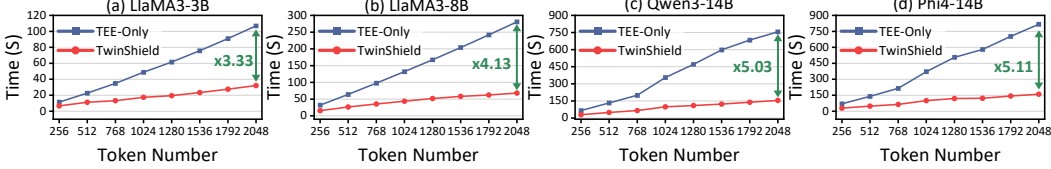

Figure 7: Prefilling latency comparison between TEE-only and TwinShield across four foundation models under varying token lengths. TwinShield consistently achieves multi-fold speedups, with gains increasing alongside model size and input token lengths.

We further compare TwinShield with related outsourcing methods, Slalom (Tramer & Boneh, 2018) and ShadowNet (Sun et al., 2023), on matrix multiplication. As shown in Figure 8, TwinShield delivers comparable or better performance while providing dual protection, whereas Slalom protects only inputs and ShadowNet protects only weights. The advantage of TwinShield stems from its combination of protocol-level outsourcing and the pipelined design OutPipe, which overlaps computation and communication to reduce idle time and achieve higher throughput.

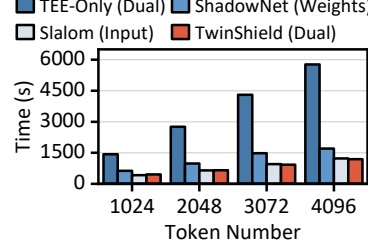

Figure 8: Compare to baselines.

**Results on long-token inputs.** Figure 7 further reports the speedup of TwinShield over the TEE-only baseline across different token lengths. The relative gain grows with input length; for instance, on Phi4-14B the speedup increases from $2.42\times$ at shorter inputs to $5.11\times$ at longer ones. This trend is consistent with our complexity analysis: in the TEE-only baseline, the enclave must perform the full $\mathcal{O}(mnp)$ multiplications, whereas TwinShield offloads these $\mathcal{O}(mnp)$ operations to the accelerator and leaves only $\mathcal{O}(mn + mp)$ lightweight scaling and additions in the TEE. As the token length $p$ increases, the gap between these complexities widens, producing larger speedups.

**Model Performance.** Our outsourcing protocols do not introduce performance drops, since we only offload heavy matrix multiplication to accelerators and the correctness is verified inside the

TEE. Thus, the model performance remains identical to the TEE-only baseline. The only source of performance degradation comes from quantization. We therefore measure perplexity (PPL) using the Wikitext (Merity et al., 2016) dataset to quantify this effect. As shown in Table 3, quantization introduces only marginal increases in PPL, while preserving the performance of FMs and enabling efficient secure execution.

**Evaluation on vision language models** To demonstrate the generality of our method across transformer-based FMs, we further test it on vision-language models (VLMs). Compared to LLMs, VLMs only introduce additional visual tokens while the overall processing pipeline remains identical. We evaluate our approach on Qwen2.5-VL-7B and Pixtral-12B, measuring the runtime for processing image inputs with prompts. Specifically, the input images all have the resolution of $960 \times 619$, which is tokenized into $805$ and $2378$ tokens. On the other hand, the input text is "Please describe this image in detail", which is tokenized into $16$ and $11$ tokens. The inference time and speedups are shown in Table 2.



Table 2: Time and speedup on VLMs.

|  | T-Vision | T-Text | TEE-Only | TwinShield |
|---|---|---|---|---|
| Qwen2.5VL-7B | 805 | 16 | 191.3 s
**1.00×** | 61.2 s
**3.13×** |
| Pixtral-12B | 2378 | 11 | 587.1 s
**1.00×** | 183.9 s
**3.19×** |

Table 3: Quantization effect on Perplexity.

|  | Original | Quantized |
|---|---|---|
| LLaMA3-3B | 10.27 | 10.63 |
| LLaMA3-8B | 7.14 | 7.84 |
| Qwen3-14B | 8.42 | 8.67 |
| Phi4-14B | 6.31 | 6.40 |



## 6.2 ABLATION STUDY AND BENCHMARK

**Ablation study on the effectiveness of proposed techniques.** Figure 9 (a) evaluates the contribution of different components of TwinShield on the Llama3-8B model. By outsourcing all multiplications through OutAttnMult and OutLinearMult, TwinShield achieves $2.71\times$ inference speedup. Building on this, the system-level optimization OutPipe further enhances the speedup to $4.13\times$ by overlapping TEE preparation, communication, and accelerator computation. This overlap eliminates the idle periods inherent in the serial workflow, thereby improving utilization on both the TEE and the accelerator and delivering substantial end-to-end performance gains.

**Performance Evaluation on Micro-Benchmark.** To isolate the effect of the protocol itself, we benchmark the matrix multiplications in `Attention` independently. Figure 9 (b) shows that OutAttnMult accelerates these operations, and with OutPipe reduces latency by $2.4\times$ at 2048 tokens. For linear layers, OutLinearMult with OutPipe achieves a $4.2\times$ reduction at the same length.

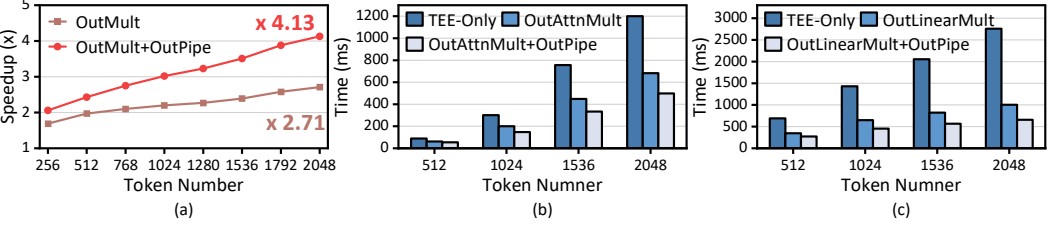

Figure 9: (a) End-to-end improvements with different techniques. OutMult includes OutAttnMult and OutLinearMult. (b) Latency of Attention Multiplication. (c) Latency of Linear Multiplication.

## 7 CONCLUSION

We propose TwinShield, a framework for secure and efficient foundation model inference that unifies TEEs with crypto-protected accelerators. By introducing OutAttnMult, OutLinearMult, and the pipelined scheme OutPipe, TwinShield achieves dual protection of inputs and models while enabling lightweight integrity verification. Experiments across multiple large-scale models show up to $5.03\times$ speedup over TEE-only execution, with ablation and micro-benchmarks confirming the complementary benefits of protocol- and system-level optimizations. These results demonstrate that TwinShield effectively bridges the gap between security and efficiency, offering a practical path toward trustworthy Foundation Model-as-a-Service.

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

APPENDIX

## A   THE USE OF LARGE LANGUAGE MODELS (LLMs)

The authors used ChatGPT and Grammarly to check and correct any typos and grammatical errors.

## B   ETHICS STATEMENT

This work focuses on improving the security and efficiency of foundation model inference by combining TEEs with cryptographic protocols. Our study does not involve human or animal subjects, nor does it require the collection of personal or sensitive data. The evaluation uses publicly available pretrained models and datasets, and no private or proprietary datasets are disclosed. We believe our method enhances privacy protection by safeguarding both user data and model confidentiality in cloud inference. The research complies with the ICLR Code of Ethics, and we are not aware of any ethical concerns or potential harms arising from this work.

## C   REPRODUCIBILITY STATEMENT

We have taken multiple steps to ensure the reproducibility of our work. The code is publicly available at https://anonymous.4open.science/r/Twinshield, together with a README file that includes instructions for installation, configuration, and execution of experiments.

## D   SECURITY ANALYSIS OF OutAttnMult

### D.1   ANALYSIS TAKEAWAY

In the outsourcing protocol in Figure 3, data inside the TEE is protected, while data processed by the accelerator may be observed by adversaries. To prevent attackers from inferring the original $Q$, the TEE constructs a masked representation $\widetilde{Q}$ by (i) adding a random mask $R_Q$, (ii) permuting $Q + R_Q$ together with $AR_Q$ under secret permutation indices, and (iii) scaling rows with a private diagonal matrix $A$.

From the adversary's perspective, recovering $Q$ from $\widetilde{Q}$ requires solving three layers of uncertainty:

1. Combination: choosing which $m$ out of the $2m$ rows correspond to the true $Q + R_Q$ block, contributing $\binom{2m}{m}$ possibilities.

2. Permutation: recovering the correct order of these $m$ rows, contributing $m!$ possibilities.

3. Scaling: guessing the diagonal scaling applied to each row, with each entry selected from the finite field $\mathbb{F}$, contributing $|\mathbb{F}|^m$ possibilities.

Putting these together, the adversary's search space is

$$\binom{2m}{m} \cdot m! \cdot |\mathbb{F}|^m,$$

and thus the security level is quantified as

$$\log_2\left(\binom{2m}{m} m! \, |\mathbb{F}|^m\right),$$

where $2m$ denotes the total number of rows in $\widetilde{Q}$ and $|\mathbb{F}|$ is the size of the finite field. Here, $\binom{2m}{m}$ captures the row combination, $m!$ the permutation order, and $|\mathbb{F}|^m$ the design space of the diagonal matrix $A$.

For a typical FM setting with input length $m = 512$ and 8-bit scalars (i.e., $|\mathbb{F}| = 2^8$), this evaluates to roughly 8,990 bits of security, which is far beyond the standard 128/256-bit levels.

## D.2 FEASIBLE SET CONSTRUCTION

We formalize the feasible set $\mathcal{F}(\hat{Z})$ of all possible pre-images $Z$ of a transformed and shuffled matrix $\hat{Z} \in \mathbb{R}^{t \times d}$ (here $Z$ represents $Q$, $K$, or $W$). Let $n$ be the number of original rows, $t = |\hat{Z}|$ the total number of rows observed, and $m = t - n$ the number of mask rows. Denote $[t] = \{1, \ldots, t\}$ and let $\mathcal{D}$ be the set of admissible non-singular diagonal scaling matrices used by the transformation.

**Assumption.** Rows in $\hat{Z}$ are generated by concatenating the original and mask rows, applying a common right diagonal scaling $D \in \mathcal{D}$, followed by a row permutation; i.e.,

$$\hat{Z} = \Pi\,[Z; R]D,$$

where $R = \begin{bmatrix} r_1^\top \\ \vdots \\ r_m^\top \end{bmatrix}$ and $\Pi$ permute $t$ rows.

1. (**Candidate mask index sets**) For any $\Omega \subseteq [t]$ with $|\Omega| = m$, define the candidate mask set
$$\Phi_\Omega = \{\hat{z}_j^\top \mid j \in \Omega\}, \qquad C_\Omega = [t] \setminus \Omega.$$
Here $\hat{z}_j^\top$ denotes the $j$-th row of $\hat{Z}$.

2. (**Candidate original rows**) Choose any injective selection $\psi : [n] \hookrightarrow C_\Omega$ and set $\overline{Z} = [\overline{z}_1^\top; \ldots; \overline{z}_n^\top]$ with $\overline{z}_i^\top = \hat{z}_{\psi(i)}^\top$.

3. (**Candidate ordering**) For any permutation $\sigma$ on $[n]$, form $\overline{Z}_\sigma = [\overline{z}_{\sigma(1)}^\top; \ldots; \overline{z}_{\sigma(n)}^\top]$.

4. (**Per-row feasible pre-images**) For each $i \in [n]$, define
$$\mathcal{F}_{\Omega,\psi,\sigma}^i(\hat{Z}) = \left\{ z^\top \in \mathbb{R}^d \mid \exists D \in \mathcal{D}, \exists \hat{z}'^\top \in \Phi_\Omega \text{ s.t. } z^\top = (\overline{z}_{\sigma(i)}^\top - \hat{z}'^\top)D^{-1} \right\}.$$

5. (**Matrix-level feasible set**) The feasible set corresponding to $(\Omega, \psi, \sigma)$ is the Cartesian product
$$\mathcal{F}_{\Omega,\psi,\sigma}(\hat{Z}) = \prod_{i=1}^{n} \mathcal{F}_{\Omega,\psi,\sigma}^i(\hat{Z}),$$
and the global feasible set is
$$\mathcal{F}(\hat{Z}) = \bigcup_{\Omega \in \binom{[t]}{m}} \bigcup_{\psi} \bigcup_{\sigma} \mathcal{F}_{\Omega,\psi,\sigma}(\hat{Z}).$$

Let the obfuscation ratio be $r := m/t$. Under fixed $n$ and a fixed $\mathcal{D}$, increasing $r$ (equivalently $t$) strictly enlarges the index-family $\binom{[t]}{m}$ and hence cannot decrease $\mathcal{F}(\hat{Z})$ (monotonicity).

## D.3 THEORETICAL GUARANTEE

**Assumption (Obfuscation Model).** All computations are carried out over a large finite field $\mathbb{F}_p$ (via fixed-point quantization). TEE samples secret diagonal scalings $A, B$, secret permutations $\lambda_Q, \lambda_K$, and fresh random mask blocks $R_Q, R_K$ uniformly at random; none are revealed to the GPU. The embedded, permuted inputs are

$$\widetilde{Q} = \text{perm}\big(Q + R_Q,\ AR_Q;\ \lambda_Q\big), \qquad \widetilde{K^T} = \text{perm}\big(K^T + R_K^T,\ R_K^T B;\ \lambda_K\big),$$

where $\text{perm}(\cdot, \cdot; \lambda)$ concatenates the two blocks along the embedding dimension and applies permutation $\lambda$.

**Theorem 1 (Indistinguishability).** For a $\widetilde{QK^T}$ computation and GPU view

$$\text{View}_{\text{GPU}} = (\widetilde{Q}, \widetilde{K^T}, \widetilde{QK^T}, (QK^T)'),$$

where $(QK^T)'$ is freshly re-masked inside the TEE, we have

$$\forall (Q_i, Q_j) \in \mathcal{F}(\widetilde{Q})^2: \quad \Pr[Q = Q_i \mid \text{View}_{\text{GPU}}] = \Pr[Q = Q_j \mid \text{View}_{\text{GPU}}], \tag{1}$$

$$\forall (K_i^T, K_j^T) \in \mathcal{F}(\widetilde{K^T})^2: \quad \Pr[K^T = K_i^T \mid \text{View}_{\text{GPU}}] = \Pr[K^T = K_j^T \mid \text{View}_{\text{GPU}}]. \tag{2}$$

*Proof sketch.*

- **Lemma 1.** The attacker cannot recover $QK^T$ from $(\widetilde{QK^T}, (QK^T)')$ because the TEE inverts the embeddings with secret $(A, B, \lambda_Q, \lambda_K)$ and then applies fresh re-masking; these are unknown to the GPU.

- **Lemma 2.** Masked vectors from $R_Q, R_K$ are indistinguishable from transformed data vectors: over $\mathbb{F}_p$, additive masking with fresh uniform vectors (scaled by secret diagonals) is a one-time pad; permutation hides positions.

Together these lemmas imply uniform posterior distributions over feasible sets, proving equation 1–equation 2.

### D.4 BOUNDING THE ADVERSARY'S SUCCESS PROBABILITY

The chance of exactly recovering $(Q, K^T)$ is bounded by the feasible-set sizes:

$$\Pr[\text{Correct}] \leq \frac{1}{|\mathcal{F}(\widetilde{Q})| \cdot |\mathcal{F}(\widetilde{K^T})|}. \tag{3}$$

Conservative lower bounds are

$$|\mathcal{F}(\widetilde{Q})| \geq \binom{t_Q}{n_Q} n_Q! \, (p-1)^{\kappa_Q}, \qquad |\mathcal{F}(\widetilde{K^T})| \geq \binom{t_K}{n_K} n_K! \, (p-1)^{\kappa_K}, \tag{4}$$

yielding

$$\Pr[\text{Correct}] \leq \left[ \binom{t_Q}{n_Q} n_Q! \, (p-1)^{\kappa_Q} \right]^{-1} \cdot \left[ \binom{t_K}{n_K} n_K! \, (p-1)^{\kappa_K} \right]^{-1}. \tag{5}$$

### D.5 DISCUSSION

Increasing the obfuscation ratio $r = m/t$ enlarges feasible sets, thereby lowering the adversary's success probability. Each round uses fresh masks $(R_Q, R_K)$ (and may refresh $A, B, \lambda_Q, \lambda_K$), ensuring independence across rounds and preventing cumulative leakage.

## E SECURITY ANALYSIS OF OutLinearMult

### E.1 THEORETICAL GUARANTEE

**Assumption (Obfuscation Model).** All computations take place over $\mathbb{F}_p$. The TEE samples: (i) a secret row-wise diagonal scaling $C = \text{diag}(c_1, \ldots, c_{d_{\text{out}}})$; (ii) fresh mask rows $R_W$; and (iii) a secret row permutation $\Pi$. These are never revealed to the accelerator. The transformed weights and masked inputs are

$$\widetilde{W} = \Pi \begin{bmatrix} W + R_W \\ C R_W \end{bmatrix}, \qquad \widetilde{X} = X + R_X,$$

with $R_X$ uniform. The accelerator computes $\widetilde{Y} = \widetilde{W}\widetilde{X}$, while the TEE recovers $WX$ using $(C, \Pi)$ and offline correction $WR_X$.

**Adversary's view.** The accelerator observes

$$\text{View}_{\text{lin}} = (\widetilde{W}, \widetilde{X}, \widetilde{W}\widetilde{X}, (WX)^\diamond),$$

where $(WX)^\diamond$ is freshly re-masked inside the TEE.

**Theorem 3 (Indistinguishability).** Conditioned on $\mathrm{View}_{\mathrm{lin}}$, all $W' \in \mathcal{F}(\widetilde{W})$ are equally likely:

$$\forall (W', W'') \in \mathcal{F}(\widetilde{W})^2 : \quad \Pr[W = W' \mid \mathrm{View}_{\mathrm{lin}}] = \Pr[W = W'' \mid \mathrm{View}_{\mathrm{lin}}]. \tag{6}$$

*Proof sketch.*

- **Lemma 3.** Since $\widetilde{X} = X + R_X$ with $R_X$ uniform, $\widetilde{X}$ is information-theoretically independent of $W$.
- **Lemma 4.** The TEE uses $(C, \Pi)$ and $W R_X$ to obtain $WX$, then re-masks it freshly. With $(C, \Pi, R_W, R_X)$ secret, the pair $(\widetilde{W}\widetilde{X}, (WX)^\diamond)$ leaks nothing about $W$.

Thus $\mathrm{View}_{\mathrm{lin}}$ is invariant across feasible pre-images, proving equation 6.

### E.2  BOUNDING THE ADVERSARY'S SUCCESS PROBABILITY

The attacker's chance of reconstructing $W$ exactly is bounded by

$$\Pr[\mathrm{Correct}] \leq \frac{1}{|\mathcal{F}(\widetilde{W})|}. \tag{7}$$

A conservative lower bound is

$$|\mathcal{F}(\widetilde{W})| \geq \binom{t}{d_{\mathrm{out}}} d_{\mathrm{out}}! \, (p-1)^\kappa, \tag{8}$$

where $t = d_{\mathrm{out}} + m$ is the number of real+mask rows and $\kappa$ the independent scales in $C$. Hence

$$\Pr[\mathrm{Correct}] \leq \left[ \binom{t}{d_{\mathrm{out}}} d_{\mathrm{out}}! \, (p-1)^\kappa \right]^{-1}. \tag{9}$$

### E.3  DISCUSSION

Monotonicity holds: increasing $r = m/t$ enlarges feasible sets and decreases the adversary's success probability. Each round uses fresh $(R_X, R_W)$ (and may refresh $(C, \Pi)$), ensuring independence across layers and preventing cumulative leakage.

## F  BACKGROUND AND RELATED WORK

### F.1  TRANSFORMERS

Transformer architecture consists of an embedding layer and consecutive transformer layers. Every transformer layer is a composition of a multi-head self-attention (MHA) module, a feed-forward (FFN) module, two normalization modules and residual connections. The input data is transformed into a token sequence through the embedding layer with positional encoding. The input token sequence can be uniformly denoted as $X_e \in \mathbb{R}^{N \times D}$, where $N$ is the number of tokens and $D$ is the embedding dimension. We describe the main computation blocks in Transformers below.

**Additive Linear Operations.** The additive linear operations in Transformers are mainly the linear layers, where the output features are computed by multiplying the input features with weight matrices. In the Attention module, given the input tokens $X_e \in \mathbb{R}^{N \times D}$, the output $Q, K, V \in \mathbb{R}^{N \times D}$ are computed by multiplying $X_e$ with three weight matrices $W_q, W_k, W_v \in \mathbb{R}^{D \times D}$:

$$Q = X_e W_q, \quad K = X_e W_k, \quad V = X_e W_v. \tag{10}$$

Similarly, in the Feed Forward module, the embedding $X_e \in \mathbb{R}^{N \times D}$ is multiplied by two weight matrices $W_1, W_2 \in \mathbb{R}^{D' \times D}$:

$$FeedForward(X_e) = Act(X_e \cdot W_1^T) \cdot W_2 \tag{11}$$

Existing techniques Tramer & Boneh (2018); Hashemi et al. (2021) can securely outsource these additive linear operations while protecting the input data $X_e$. However, they cannot protect the weight matrices $W$, which remain exposed to the untrusted accelerator.

**Multiplicative Attention Operations.** There are massive multiplicative linear operations in Transformers which cannot be outsourced via prior methods. The primary multiplicative attention operations are computing the attention map and attention output in the Attention module:

$$Attention(Q, K, V) = SoftMax(QK^T/\sqrt{d_h})V \tag{12}$$

The multiplicative operations (e.g., $Q \cdot K^T$) are fundamentally different from the additive linear operations (e.g., $Q = X_e \cdot W_q$). This is because in the multiplicative operations, neither operand is a constant matrix. As a result, the multiplicative operations cannot be securely outsourced via existing techniques. We refer to the matrix multiplication between $Q$ and $K$, and between the attention map and $V$ as attention matrix multiplication.

The multiplicative linear operations such as $Q \cdot K^T$ are computed independently across multiple attention heads. For MHA with $H$ heads, the multi-head attention is computed as:

$$MHA(Q, K, V) = Concat(head_1, ..., head_H)W_O \tag{13}$$

where $Concat(\cdot)$ is the concatenation operation,

$$head_i = Attention(XW_q^i, XW_k^i, XW_v^i) \tag{14}$$

and $W_O \in \mathbb{R}^{Hd_h \times D}$ is a weight matrix to map features in all heads to the output dimension. The MHA is the key mechanism in the Transformers and also the performance bottleneck. However, existing works cannot securely outsource the heavy computation in the multiplicative linear operations within the MHA module.

### F.2 TRUSTED EXECUTION ENVIRONMENTS (TEES)

TEEs like Intel SGX (Intel SGX) provide a secure environment where data confidentiality and, in some cases, computation integrity are ensured by hardware. Intel SGX specifically safeguards the confidentiality and integrity by isolating data and code within an enclave, shielded from external elements including the operating system, hypervisor, and hardware devices on the system bus. This isolation involves a dedicated memory region, the Processor Reserved Memory (PRM), managed by SGX-enabled CPUs. Here, the Enclave Page Cache (EPC) stores enclave data and code in 4 KB pages, accessible only through specific CPU instructions. This setup prevents unauthorized access to the EPC, maintaining a secure environment for sensitive computations. SGX also supports remote attestation, allowing remote verification of an enclave's integrity through cryptographic proofs. These features have inspired research into running deep learning inference entirely within CPU TEEs to protect data and model confidentiality (Hanzlik et al., 2021). However, the high computational and memory demands of deep learning models make CPU TEE inference inefficient, motivating the use of accelerators such as GPUs, TPUs, and ASICs to improve performance.

**TEE with AI Accelerators.** Although some high-end accelerators (e.g., NVIDIA H100 (Choquette, 2023)) have begun to support TEE capabilities, enabling secure computation directly on the accelerator remains impractical in most real-world deployments. This is due to two key factors: firstly, TEE-enabled accelerators are rare and expensive, while many emerging and legacy GPUs (such as GTX series and A100) currently deployed in data centers remain in use and are likely to persist for years. Secondly, the growing heterogeneity of hardware accelerators (e.g., GPUs, TPUs, and FPGAs) introduces vendor incompatibilities and the complexity of cross-device TEE protocols, making a unified TEE solution across diverse devices infeasible;

For accelerators without native TEE support, a more promising approach treats the CPU TEE as the root of trust, and considers the accelerator as an untrusted but controlled extension of the TEE. In this design, sensitive data is decrypted and processed inside the CPU TEE, which enforces strict isolation and integrity guarantees, while delegating computationally intensive linear operations to the untrusted accelerator through carefully controlled and isolated interfaces. These interfaces may include exclusive device assignment, core pinning, and secure memory buffer management to minimize the risk of confidential data leakage. To further mitigate the risk of exposing plaintext data during offloading, cryptographic techniques such as secret sharing are integrated, allowing the CPU TEE to securely partition computations, outsource them to the accelerator, and verify the correctness of results upon return. However, this approach remains limited in its ability to support attention mechanisms in Transformer-based models, which are critical to modern deep learning workloads.

Therefore, it is imperative to design advanced secure outsourcing schemes that preserve the CPU TEE as the single root of trust while efficiently leveraging heterogeneous and vendor-diverse untrusted accelerators to fully exploit their performance potential without compromising security.

### F.3 SECRET SHARING FOR DATA CONFIDENTIALITY

Secret Sharing Cramer et al. (2015); Demmler et al. (2015) is a cryptographic primitive that allows multiple parties to compute a function over their inputs while keeping them private. All our algorithms are built on a two-party secret sharing over the field $\mathbb{F}_p$, where $p$ is a prime number indicating field size. In a two-party secret sharing, a secret $x$ is split into two shares by random sampling $\langle x \rangle_0, \langle x \rangle_1 \in \mathbb{F}_p$, such that $x = \langle x \rangle_0 + \langle x \rangle_1 \mod \mathbb{F}_p$. Secret sharing offers a strong security guarantee that, given a share $\langle x \rangle_0$ or $\langle x \rangle_1$, the value of the original $x$ is hidden, i.e., either party can reconstruct the value of $x$ with negligible possibility Cramer et al. (2015). In the setting of TEE-based confidential inference, the value $x$ can be split by a randomness $r \in \mathbb{F}_p$ chosen by the TEEs, such that the two shares are $\langle x \rangle_0 = r$ and $\langle x \rangle_1 = x - r$, respectively. Prior works Tramer & Boneh (2018); Sun et al. (2023) employ secret sharing to provide privacy guarantees when outsourcing additive linear operations with constant weights $w$. Yet, existing outsourcing schemes cannot be extended to multiplicative attention operations where both operands are variables, such as $Q$ and $K$, as it is impossible to precompute multiplication between $r$ and either $Q$ or $K$. Even for additive linear operations, they only protect the input data $x$, leaving the weight matrix $W$ exposed to the untrusted accelerator.

### F.4 COMPUTATION VERIFICATION FOR INTEGRITY

The verification algorithm enables a client to assert the correctness of computations performed by a server. Within the landscape of TEEs, where computations are outsourced to high-performance untrusted devices such as GPU, ensuring the integrity of these operations is paramount. Soter (Shen et al., 2022) introduces a "fingerprint" matrix method for integrity checks by the TEEs, which, however, may be vulnerable to targeted attacks. Additionally, recent research Wei et al. (2023) suggests a sampling-based verification by the TEEs to compare against GPU outputs, facing limitations in detecting selective manipulations without extensive sampling. Freivalds' algorithm Freivalds (1977), referenced in Tramer & Boneh (2018); Sun et al. (2023); Hashemi et al. (2021), provides an efficient mechanism for verifying matrix multiplications of the form $AB = C$. The algorithm commences by generating a random vector $r$, followed by the TEEs computing the products $B \cdot r$ and $C \cdot r$. The next step involves multiplying $A$ with $B \cdot r$, and comparing this outcome to $C \cdot r$. A discrepancy between these products indicates a failure of $AB$ to equal $C$, whereas a match suggests a probable equality between $AB$ and $C$. Employing this method, the TEEs are able to perform a verification of $\mathcal{O}(n^3)$ matrix multiplication complexity using a more efficient $\mathcal{O}(n^2)$ vector-matrix multiplication operation, thereby enhancing the verification efficiency within the TEEs.

