# OpenReview forum: "TwinShield: Secure Foundation Model Execution by Unifying TEEs and Crypto-protected Accelerators"
_ICLR.cc/2026/Conference — Submitted to ICLR 2026_

### Official Review · Reviewer_Ry8i · 2025-11-01

**Soundness:** 2
**Presentation:** 3
**Contribution:** 2
**Rating:** 4
**Confidence:** 3

**Summary:**

TwinShield is a secure foundation model inference framework that combines Trusted Execution Environments (TEEs) with untrusted accelerators to protect both user data and model parameters during cloud-based execution. It introduces two cryptographic outsourcing protocols, OutAttnMult for attention layers and OutLinearMult for linear layers, that securely offload heavy computations, and a pipelined scheme (OutPipe) that overlaps TEE processing and GPU computation to reduce communication overhead. Experiments on large language and vision-language models show 3.3×–5.1× speedups over TEE-only baselines while maintaining model accuracy and verifiable integrity. TwinShield demonstrates that secure and efficient foundation model inference is achievable without sacrificing confidentiality or performance.

**Strengths:**

- The paper addresses an underexplored problem by simultaneously protecting sensitive user data and proprietary model parameters during cloud-based foundation model inference.

- The authors evaluate TwinShield on a diverse set of large-scale models, including both language and vision-language models, and provide comprehensive analyses of computational cost and theoretical security.

**Weaknesses:**

- The paper does not sufficiently explain why GPU-based TEE solutions (e.g., NVIDIA H100 Confidential Mode) fail to provide the same confidentiality and integrity guarantees as CPU-based TEEs, making the solution less attractive.
- The related work discussion omits lightweight cryptographic or obfuscation-based alternatives, such as Secure Transformer Inference Protocol [1] and TransLinkGuard [2]. Although [1] focuses on multi-party computation without TEEs and [2] primarily protects model weights, both employ permutation-based obfuscation strategies that could achieve comparable confidentiality goals with less complexity. Including a direct comparison with these techniques would clarify the relative trade-offs between cryptographic cost, trust assumptions, and performance efficiency.

[1] Yuan, Mu et al. “Secure Transformer Inference Protocol.” (2023).

[2] Li, Qinfeng et al. “TransLinkGuard: Safeguarding Transformer Models Against Model Stealing in Edge Deployment.” (2024).

**Questions:**

- The performance evaluation in Figure 7 focuses on prefilling latency. How does TwinShield's performance, particularly the OutPipe pipelining, hold up during the autoregressive decoding phase?

---

### Official Review · Reviewer_9Fty · 2025-11-02

**Soundness:** 2
**Presentation:** 3
**Contribution:** 2
**Rating:** 2
**Confidence:** 4

**Summary:**

This paper proposes a framework for secure inference of transformer-based foundation models by combining trusted CPU processors with untrusted accelerators like graphics processing units. The approach addresses two key challenges: enabling dual protection of both user data and model parameters during inference, and mitigating communication overhead between processors and accelerators at scale. The authors design three protocols: OutAttnMult for securely outsourcing attention computations, OutLinearMult for linear layer operations, and OutPipe for pipelining that overlaps processing preparation, communication, and acceleration to reduce idle time. The core cryptographic technique uses scale-then-permute masking with diagonal scaling matrices and permutations to obfuscate operands. The framework employs fixed-point quantization and uses Freivalds algorithm for integrity verification. Evaluation on multiple large language models and vision-language models demonstrates that the system offloads approximately 87 percent of computations to accelerators, achieving 3.3 to 5.1 fold speedups compared to processor-only baselines, with quantization causing only marginal perplexity increases.

**Strengths:**

- Addresses practical dual protection need: Unlike prior systems protecting only data or only models, this paper targets simultaneous confidentiality for both during cloud inference, a real deployment requirement for foundation model services
- System-level optimization through pipelining: The proposed pipelined design overlaps processor preparation, data transfer, and acceleration computation, achieving measured 52.4 percent improvement by exploiting attention head independence. This practical contribution delivers measurable throughput gains.
- Comprehensive evaluation on large models: Testing spans models up to 14 billion parameters including both language and vision-language architectures, with detailed ablation studies demonstrating individual component contributions
- Implementation accessibility: The authors provide code built on established frameworks (ggml, llama.cpp), supporting reproducibility

**Weaknesses:**

- Recycled cryptographic techniques: The scale-then-permute strategy and diagonal matrix masking derive directly from prior systems designed for convolutional neural networks. Adaptation to attention represents engineering contribution rather than novel cryptographic innovation. The distinction between this paper and prior work is primarily architectural rather than methodological.
- Missing critical security evaluation: The paper omits comparison with "No Privacy Left Outside" (S&P 2024), which demonstrates that processor-shielded partitioning approaches are vulnerable to model extraction attacks. This represents a fundamental threat to the security model. Related recent work such as HyperTheft shows ciphertext side channels can extract model weights from processor-shielded systems, directly contradicting claimed security guarantees.
- Incomplete related work coverage: Missing substantial recent secure transformer inference work including cryptographic multiparty computation approaches (Iron, BOLT, Nimbus, STIP), homomorphic encryption methods (THE-X, CipherFormer), and hybrid processor-accelerator designs (THEF, SecureInfer). This gap undermines novelty claims.

**Questions:**

- Security against recent attacks: How does this paper defend against model extraction demonstrated in "No Privacy Left Outside" (S&P 2024) and HyperTheft? These papers show processor-shielded partitioning is fundamentally vulnerable. Please provide: empirical evaluation against these attack methods, information leakage analysis from permutation patterns, and security guarantee comparison with prior systems subsequently broken.
- Comprehensive baseline comparisons: Comparisons are limited to processor-only execution, Slalom, and ShadowNet. Please compare with recent transformer-specific secure inference systems: Nimbus, BOLT, STIP, and THEF. Provide performance breakdown showing whether advantages derive from cryptographic protocols, system optimization, or quantization. Compare security-efficiency tradeoffs against cryptographic multiparty computation and homomorphic encryption approaches.
- Practical evaluation of: Detailed communication cost analysis (bytes transferred, protocol rounds, comparison against related work reporting these metrics), Freivalds verification overhead (actual repetition count in practice and resulting integrity guarantee)
- What are specific technical advantages over THEF, PrivMLLM, and SecureInfer?

---

### Official Review · Reviewer_Vks3 · 2025-11-03

**Soundness:** 3
**Presentation:** 3
**Contribution:** 2
**Rating:** 2
**Confidence:** 4

**Summary:**

The paper proposes to use existing trusted execution environments for securely evaluating transformer models. The main novelty is to outsource some part of the computation to an untrusted GPU.

**Strengths:**

Some improvements for outsourcing some computation for TEEs.

Experimental evaluation on small transformer models.

**Weaknesses:**

I am not fully convinced that using the B factor is secure if it remains fixed throughout the outsourcing.

The novelty of the proposed approach also appears somewhat limited compared to existing work.

Furthermore, I disagree with the paper’s argument that using GPU based TEEs—such as the NVIDIA H100 for secure GPU outsourcing—is problematic. In fact, this approach appears to be aligned with current industry trends and represents a practical direction for secure computation.

**Questions:**

None.

---

### Official Review · Reviewer_ckdE · 2025-11-03

**Soundness:** 2
**Presentation:** 3
**Contribution:** 2
**Rating:** 2
**Confidence:** 3

**Summary:**

This paper proposes a scheme, named TwinShield, to securely offload attention computation in LLMs and VLMs from a trusted CPU TEE to an untrusted accelerator. Similar approaches have been proposed and studied before for MLPs/CNNs when only inputs (but not model weights) need to be protected. TwinShield uses the masked product approach from the previous work, but protects additional product computation needed for attention calculation through secret scaling and a secret column-wise permutation (OutAttnMult, OutLinearMult). For performance, the paper also proposes to pipeline the computation and communication between the CPU TEE and the accelerator (outPipe). The experimental results on an Intel SGX CPU TEE with an NVIDIA A40 GPU show 3-5x speedups compared to the SGX TEE-only case for LLMs (Llama3, Qwen3, Phi4) and VLMs.

**Strengths:**

The paper points out the limitations in previous secure computation outsourcing schemes for ML inference in the context of Transformers, and proposes a potential solution that can enable protecting both inputs and model weights. If the scheme is truly secure and efficient, it advances the state-of-the-art in secure inference outsourcing.

The proposed scheme is prototyped on a commercial TEE system, and the performance was evaluated for multiple LLMs and VLMs. The performance results look promising compared to the CPU TEE-only solution based on Intel SGX.

**Weaknesses:**

The following summarizes the main questions that the paper needs to address to more fully justify the security and the effectiveness of the proposed scheme.

1. Security
While the proposed scheme may prevent the substitution attack discussed in the paper, the current security analysis is insufficient to truly prove the security. The paper needs a formal security proof.

The security analysis (in Section 4.1) simply states that Q+R_Q forms a one-time pad and is indistinguishable from A*R_Q. However, it is not clear if a simple scalar matrix (A) can truly hide R_Q and completely remove the correlation between Q+R_Q and A*R_Q. As an example, if an input (X) has mostly zero elements, Q+R_Q will be close to R_Q, and it appears that guessing A will be easier with Q+R_Q. The security guarantee should hold for all possible inputs and weight values and even when an adversary may know some of their values.

2. Evaluation
The paper only shows the performance improvement over the baseline using Intel SGX, which had limited secure memory capacity. However, practical deployments of CPU TEEs are mostly based on VM-based TEEs such as Intel TDX and AMD SEV, which support much larger protected memory capacity. The performance overhead of running large ML models should be much lower for the modern CPU TEE designs. To better justify the effectiveness of the proposed scheme, the paper needs to show its benefit for VM-based TEEs.

Also, the reported latency of tens of seconds still seem quite slow. To more fully evaluate the overhead and practicality, the paper should provide performance comparisons with the GPU TEE and the baseline w/o TEE. It is also unclear how the overhead of offline computations is considered in the evaluation. Finally, it will be helpful to study both prefill and decode latencies (the current evaluation appears to only study prefill latency).

3. Motivation
The main technical arguments that the paper makes to motivate outsourcing partial inference operations to untrusted accelerators are not well supported.

The paper refers to a previous study on the Intel SGX, which had a limited secure (enclave) memory capacity, to point out running the ML model entirely in a CPU TEE is too expensive. However, more recent VM-based CPU TEEs such as Intel TDX and AMD SEV support much larger protected memory capacity and should have much lower overhead. In that sense, to be more compelling, the paper needs to show the performance benefits compared to these VM-based CPU TEEs.

The paper states that recent accelerator TEEs (such as NVIDIA GPU TEE) provides weaker guarantees than CPU-
based TEEs to motivate the use of untrusted accelerators. While the problem of how/if we can securely offload attention computation to untrusted accelerators is an interesting question by itself, the claim that accelerator TEEs are less secure than a CPU TEE is not justified. In fact, CPUs often rely on more complex optimizations such as speculative execution and are often shared with more untrusted software. Recent attacks on CPU TEEs such as TEE.fail show that CPU TEEs can be more easily attacked through side-channels or DRAM interposers - attacks that  do not apply to accelerators with HBMs. If one cannot trust the security of an accelerator TEE, it is not clear why it is reasonable to assume that a CPU TEE (which is more complex and exposed to more types of software and physical attacks) can be trusted.

**Questions:**

1) Can you provide a formal proof of security? It is not clear how a simple scalar matrix (A) can completely remove the correlation between Q+R_Q and A*R_Q.

2) How will the performance benefit of TwinShield change for VM-based CPU TEEs (Intel TDX, AMD SEV)? How does the TwinShield performance compare to the baseline w/ no TEE and the GPU TEE?

3) How does the paper's performance evaluation include/consider the offline computation? Recent studies (such as https://arxiv.org/pdf/2207.07177) suggest that offline overhead is important to consider for throughput.

---

### Meta-Review · Area_Chair_mEBF · 2026-01-11

**Summary:**

This paper proposes a framework for secure inference of transformer-based foundation models by combining trusted CPU processors with untrusted accelerators. Reviewers agree that the problem is relevant and that the system implementation is non-trivial, with evaluations conducted on multiple LLM and VLM architectures.

**Reviewer Concerns:**

All reviewers raise significant concerns about security rigor, novelty, and evaluation completeness. The security analysis lacks formal guarantees and does not fully demonstrate that the proposed methods fully prevent information leakage under realistic adversarial assumptions. The cryptographic techniques are largely adapted from prior work, with the main contribution being architectural integration rather than methodological novelty. The motivation relies heavily on limitations of older enclave-based TEEs, while not adequately considering modern VM-based CPU TEEs or GPU-based TEEs, whose threat models and performance characteristics may differ substantially. The evaluation is also limited.

All reviewer concerns remain unaddressed.

**Reviewer Scores:**

There is no author response, so it's likely that the reviewers won't change their ratings.

---

### Decision · Program_Chairs · 2026-01-26

Reject